# Phage Engineering for Targeted Multidrug-Resistant *Escherichia coli*

**DOI:** 10.3390/ijms24032459

**Published:** 2023-01-27

**Authors:** Jiaoyang Song, Zhengjie Liu, Qing Zhang, Yuqing Liu, Yibao Chen

**Affiliations:** Institute of Animal Science and Veterinary Medicine, Shandong Academy of Agricultural Sciences, Jinan 250000, China

**Keywords:** CRISPR/Cas9, engineering phage, phage tail fibers, multidrug resistant bacteria

## Abstract

The lytic bacteriophages have potential application value in the treatment of bacterial infections. However, the narrow host spectrum of these phages limits their range of clinical application. Here, we demonstrate the use of scarless Cas9-assisted recombination (no-SCAR) gene-editing technology to regulate phage–host range. We used phage PHB20 as the scaffold to create agents targeting different multidrug-resistant *Escherichia coli* by replacing its phage tail fiber gene (ORF40). The engineered phages were polyvalent and capable of infecting both the original host bacteria and new targets. Phage-tail fiber genes can be amplified by PCR to construct a recombinant phage PHB20 library that can deal with multidrug-resistant bacteria in the future. Our results provide a better understanding of phage–host interactions, and we describe new anti-bacterial editing methods.

## 1. Introduction

The irrational use of antibacterial agents in animal production has facilitated the occurrence of multidrug resistance in food-borne pathogens. Among the many drug-resistant bacteria, multidrug-resistant *Escherichia coli* are regarded as pathogenic bacteria that pose a major threat to public health. A large amount of research has shown that vegetables, meat, eggs, and dairy products contaminated by multidrug-resistant *E. coli* seriously threaten human health [1,2,3,4,5]. As viruses that specifically infect bacteria, bacteriophages could be used as an alternative to antibiotics in treating bacterial infections in plants, animals, and human beings [6,7,8,9,10]. Moreover, phages have a wide range of applications in the detection of bacteria, removal of bacterial biofilms, and amelioration of the bacterial pollution of food [11,12,13,14,15].

Although phages have achieved certain therapeutic effects in clinical applications, the availability of specific phages that infect certain bacteria, and the narrow host ranges of phages are a major limiting factor in their operability. Recently, phage engineering has provided a strategy for expanding the phage host spectra and the ability to kill resistant bacteria [16,17]. Using engineered phages, clinicians successfully treated a patient infected by drug-resistant *Mycobacterium abscessus* [18]. Another genetically modified *Staphylococcus aureus* phage was used to treat a murine shin infection, osteomyelitis, and soft-tissue infection [19,20]. In addition, modified phages can serve as antigen-delivery vehicles to prevent animal diseases [21,22].

In recent years, the continuous increase in multidrug-resistant bacteria has aroused the scientific communities’ interest in using phages as antibacterial agents. The number of phage particles in the biosphere is estimated to be 10^31^, more than 10 times that of their host cells [23]. Ninety-six percent of all known phages are tailed [24], and phage tails exhibit a great diversity of functions in host recognition. Although phages isolated from pure cultured bacteria have been used in treating few bacterial diseases, this application is seriously impeded by the shortage and lack of diversity of phages that can infect drug-resistant bacteria. Previous studies have shown that it is possible to expand the phage–host spectrum through phage-tail editing or artificial synthesis technologies [16,17,25,26]. However, most research focuses on genetically modifying cultivable phages. In addition, some factors, including operability, time consumption, and recombination efficiency, limit the development of gene-editing technologies. 

We have developed a universal and easy-to-use phage-engineering platform for the modification of phage PHB20, which has potential uses in biomedical applications. In this study, we focused on obtaining phage-tail fiber genes directly from an environmental water sample, and using them individually to replace the PHB20-tail fiber gene (ORF40). We found that doing so successfully expanded the host spectrum of recombinant PHB20 phages. Our data highlight the utility of engineered phage libraries as strategic solutions for dealing with multidrug-resistant bacteria.

## 2. Results and Discussion

Recently, we used the scarless Cas9 assisted recombineering (no-SCAR) system as a powerful tool for genome editing of an *E. coli* phage [25]. To test the usefulness of the system for phage engineering, we first constructed a small guide RNA (sgRNA) plasmid with different target sequences from the virulent phage PHB20 (belonging to the *Podoviridae* family), with *E. coil* BL21 as a host; PHB13 (belonging to the *Myoviridae* family), with *E. coil* O157 as a host; and SP01 (belonging to the *Siphoviridae* family), with *Salmonella* ATCC14028 as a host. A schematic overview of the phage-editing workflow is presented in Figure 1. We randomly chose 10 putative protospacer-sequences -targeting genes for each phage. A lower plating efficiency of 0 to 10^−1^ was observed for phage PHB20 (Figure 1C). Phage PHB13, for which only sgRNA-ORF106 spacers showed antiviral activity, exhibited a plaque-formation efficiency of ~10^−3^ (Figure 1C), whereas the other spacers did not induce an immune response. The discrepancies in antiviral immunity observed for different phages may be related to the spacer used; for example, phage SP01 sgRNA-ORF17, sgRNA-ORF23, sgRNA-ORF25, sgRNA-ORF29, and sgRNA-ORF35 spacers showed limited cutting efficiency in the host cell, while ORF46 spacer showed a higher cutting efficiency and a plating efficiency of ~10^−1^ (Figure 1C). There were no obvious correlations between the GC or C contents of the protospacer sequences and gene cutting efficiency. Our data showed that the no-SCAR system can be adapted for gene editing of coliphages and *Salmonella* phages.

Based on comparative genome analysis, phage PHB20 (only infects BL21 and DH5α), one of the T7-like members of the *Autographivirinae* subfamily, showed 92.8% nucleotide-sequence identity to phage T7 (GenBank no. AY264774). The tail fiber sequences, encoded by ORF40 in PHB20, are equivalent to the T7 phage *gene 17*. The tail fiber of the T7 phage contains two major domains: the N-terminal domain made of 149 amino acid residues, a highly conserved region that is believed to bind the capsid protein; and the C-terminal domain, which is variable, allowing adaption to different hosts [26]. Whole-genome comparison analysis revealed that phage PHB20 and podovirus sequences available in the NCBI databases show a high degree of nucleotide similarity and a consistent mosaic arrangement, and only genes encoding the tail fiber protein show diversity (Appendix A). Therefore, we attempted to develop a simple method to expand the phage–host spectrum by directly replacing the tail fiber gene of PHB20. Based on the nucleotide homology, we designed a pair of primers to amplify podovirus-tail fiber genes directly from environmental water samples (Appendix A). Following the operation shown in Figure 2, we collected 43 environmental water samples, and removed the bacteria. The phage particles were purified by CsCl gradient ultracentrifugation. To prepare the sample for PCR, phage genomes were extracted; phage DNA was directly extracted the from the water sample. The primers F: 5′-ATGGCTAACGTAATTAAAACCG-3′ and R: 5′-CGACTACCTTGGCACCAATCT-3′ were used in this study, and Sanger sequencing was performed. From the results, 30 different phage-tail fiber genes were obtained (Excel S1). 

We hypothesized that these phage genes encode different tail fibers that may function in host recognition, as the primary host determinant for the phage T7 is the tail fiber product of *gene* 17. The variable sequences of phage-tail fiber genes are related to their recognition of different host receptors and changes in the host range [16]. Next, the purified amplicons were cloned and swapped with phage PHB20 sequences, as illustrated in Figure 3A,B. A series of recombinant PHB20 phages were constructed, and the phage–host range was determined. We found that phage PHB20 could infect *E. coli* BL21 and did not produce plaques in a multidrug-resistant *E. coli* E22 strain, while swapping the tail fiber gene of the engineered phage PHB20(P6) not only led to it infecting the wild-type cell, but also the E22 strain (Figure 3C,D). Subsequently, we constructed 30 recombinant PHB20 phages, all of which showed potential antibacterial abilities (data not shown).

Our research indicates that phage-tail fiber genes of phage PHB20 can be swapped to expand its host spectrum. Our results showed that the engineered phages retained the biological characteristics of the original phage, and could kill multidrug-resistant host cells. The PCR method was used to directly obtain phage-tail fiber genes from environmental water samples. Subsequently, swapping these phage-tail fiber genes established an engineered phage library. Similarly, the metagenomic database contains abundant phage-tail fiber genes, which can be synthesized and swapped with endogenous genes to form the desired model phage particles. It is possible to employ recombinant phage libraries to meet the current need for new antibacterial agents in the face of a continuous increase in multidrug-resistant bacteria.

## 3. Materials and Methods

### 3.1. E. coli Growth Media

*E. coli* DH5α and BL21 (DE3) were cultured in LB broth at 37 °C. If required, antibiotics were added to the medium. Super optimal broth (SOB), instead of LB broth was used for the transformation of *E. coli* DH5α. pKDsgRNA-ack (spectacomycin-resistance gene, Addgene plasmid #62654), pCas9cr4 (chloramphenicol-resistance gene, Addgene plasmid #62655) and donor plasmids (ampicillin-resistance gene, Addgene plasmid #50004) were purchased from ThermoFisher.

### 3.2. Transformation of E. coli BL21

pCas9cr4 plasmids and sgRNA plasmids were mixed with 50 μL of competent BL21, which was transferred to a 0.2-mm electroporation cuvette and then incubated on ice for 30 min. Then, 100 μL of bacteria was added to coat an LB plate containing antibiotics (34 μg/mL chloramphenicol and 50 μg/mL spectinomycin) and incubated overnight at 30 °C. Anhydrotetracycline (100 ng/mL) and L-arabinose (50 mmol/L) were added when the bacteria reached the growth phase (OD630_nm_ = 0.4), and these were incubated at 30 °C for 30 min.

### 3.3. Phage Infection E. coli BL21

A 300-μL aliquot of strain BL21 (containing three plasmids: pCas9cr4, sgRNA, and donor plasmids) was mixed with 10^4^ to 10^5^ PFU of phage PHB20, and poured onto 6 mL of soft LB agar containing 34 μg/mL chloramphenicol, 50 μg/mL spectinomycin, 50 μg/mL ampicillin, 100 ng/mL anhydrotetracycline and 50 mmol/L L-arabinose, and incubated at 30 °C overnight. The conventional double-layer agar method was used to pick up 8–12 random phage plaques. Genomic DNA was extracted from each purified possible recombinant phage. The above detailed methods were previously described [25].

## Figures and Tables

**Figure 1 ijms-24-02459-f001:**
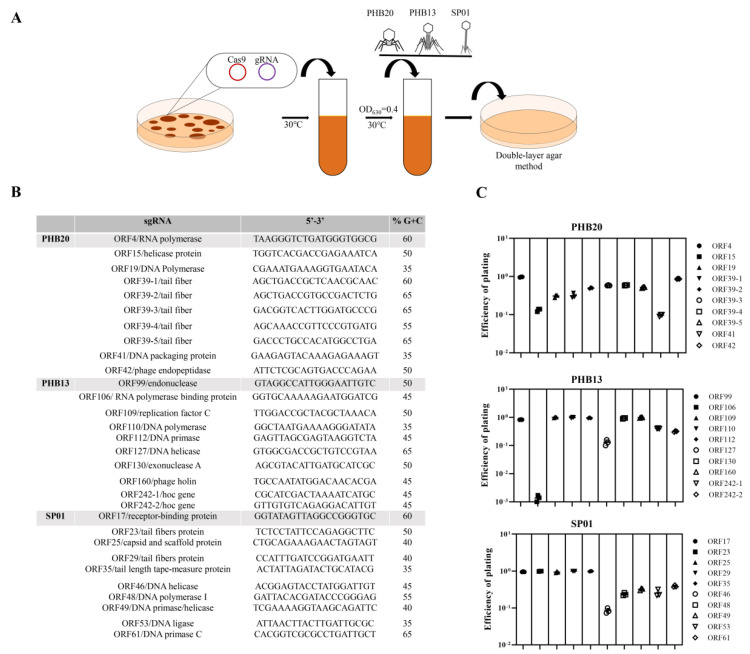
Efficiency of the no−SCAR system protection against phage infection. (**A**) Schematic overview of the no−SCAR−system−based phage genome editing. (**B**) The spacer sequences from different phages. (**C**) The efficiency of plating was determined by a plaque assay. This experiment was performed in triplicate: PHB20 (Genbank no. MN481366), PHB13 (Genbank no. MK573636), and SP01 (Genbank no. KY114934). Different symbols represent different protospacer sequences.

**Figure 2 ijms-24-02459-f002:**
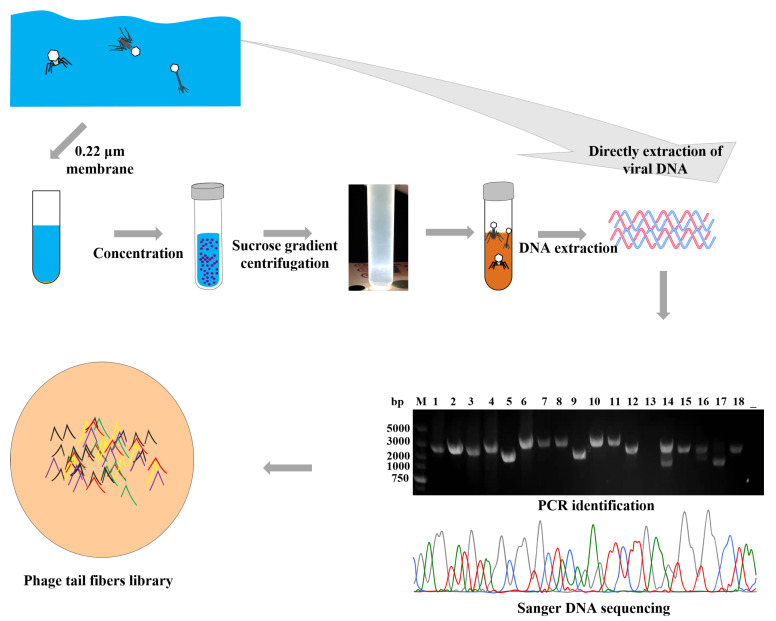
Schematic showing the collecting of phage-tail fiber genes. The phage particles were purified by CsCl gradient ultracentrifugation. Amplified PCR product from phage-tail fiber genes, and the numbers 1-18 represent the different phage-tail fiber gene fragments.

**Figure 3 ijms-24-02459-f003:**
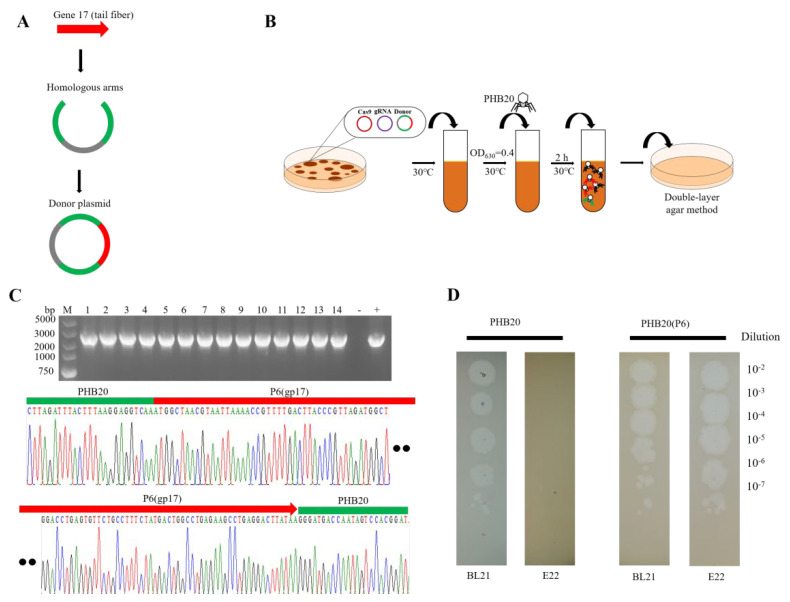
Recombinant phage PHB20. (**A**) Construction of donor plasmids. All phage-tail fiber gene templates were performed using the CloneExpress^®^ II One Step Cloning Kit. (**B**) Schematic overview of the workflow of PHB20 genome editing. (**C**) Sanger sequencing of the target genes. The numbers 1−14 represent recombinant phages. (**D**) Plaquing assay with WT−PHB20 and engineered phage PHB20(P6). Ten-fold serial dilutions of phage lysates were spotted on bacterial lawns and incubated at 37 °C for 6 h.

## Data Availability

The annotated genome sequence of PHB20, PHB13, SP01 and were submitted to the GenBank database under the accession number MN481366, MK573636, and KY114934, respectively.

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
