# Peer review of "Phage Engineering for Targeted Multidrug-Resistant Escherichia coli"

_ijms, 2023, doi:10.3390/ijms24032459_

Round 1

Reviewer 1 Report

I believe that the manuscript must be completely rewritten. We can start with the title. What that words ' The strategy of genes to obtain the source of water environment' means?

“the availability of specific phages infecting certain bacteria and the narrow host range of phages are still the major limiting factor.” May be factors? How is the availability of specific phages the main limiting factor?

“the operability … are still the reasons that limit the development of gene editing technology”???

“the CRISPR-Cas9 system is very active in the phage defense, providing the basis for Cas19-based phage gene editing in E. coli.” What is phage defense from and what is cas19?

“There was no obvious correlation between the GC or C content of the protospacer sequences and the level of antiviral immunity.” It should be? Why?

“Our research indicated that phage PHB20 can swap its phage tail fiber genes and expand the host spectrum.” Can the phage PHB20 change genes on its own?

And so on…

The methods used are not described. Reference could be made to the methods described in other works.

Author Response

We would like to express our sincere thank you comments and suggestions, which allowed us to further improve our manuscript. This manuscript has been carefully modified according to the comments. The following response is point-by-point towards the reviewers’ comments.

  1. I believe that the manuscript must be completely rewritten. We can start with the title. What that words ' The strategy of genes to obtain the source of water environment' means?

We thank the reviewer for the positive comment. We have modified accordingly. We consulted two English experts to improve our language.

  1. “the availability of specific phages infecting certain bacteria and the narrow host range of phages are still the major limiting factor.” May be factors? How is the availability of specific phages the main limiting factor?

Thanks for the suggestions. We have made corrections in the manuscript.

“Although phages have achieved certain therapeutic effects in clinical applications, the availability of specific phages that infect certain bacteria and the narrow host ranges of phages are major limiting factor in their operability.”

  1. “the operability … are still the reasons that limit the development of gene editing technology”???

Thanks for the suggestions. We have made corrections in the manuscript.

“However, most research has on genetically modifying  cultivable phages. In addition, some factors, including operability, time consumption, and recombination efficiency, limit the development of gene-editing technologies”

  1. “the CRISPR-Cas9 system is very active in the phage defense, providing the basis for Cas19-based phage gene editing in E. coli.” What is phage defense from and what is cas19?

Thanks for the suggestions. We have made corrections in the manuscript.

“Phage PHB13, for which only sgRNA-ORF106 spacers showed antiviral activity, exhibited a plaque-formation efficiency of ~10-3 (Figure 1C), whereas the other spacers did not induce an immune response.”

  1. “There was no obvious correlation between the GC or C content of the protospacer sequences and the level of antiviral immunity.” It should be? Why?

Thanks for the suggestions. We have made corrections in the manuscript.

“There were no obvious correlations between the GC or C contents of the protospacer sequences and the gene cutting efficiency.”

  1. “Our research indicated that phage PHB20 can swap its phage tail fiber genes and expand the host spectrum.” Can the phage PHB20 change genes on its own?

Thanks for the suggestions. We have made corrections in the manuscript.

“Our research indicates that phage tail fiber genes of phage PHB20 can be swapped to expand its host spectrum.”

  1. And so on…

We have made corrections in the manuscript.

  1. The methods used are not described. Reference could be made to the methods described in other works.

We have add the reference and methods in the manuscript.

Reviewer 2 Report

The manuscript: ‘The strategy of phage tail fiber genes to obtain the source of water environment: Engineering phage for targeted multidrug resistant Escherichia coli’ introduces a gene editing approach to modify narrow host range phage. It is shown that after modification of tail fiber the PHB20 phage can infect alternative host carrying multiple antibiotic resistances. There are several points I want to address and I recommend major modification prior considering for publication.

The authors have an interesting approach to fight increasing antibiotic resistance. I think the research itself has been conducted using novel and valid technique but there are some parts in the manuscript not explained well and missing essential information. I also understand it is a communication article with limited space but mainly the procedure and the justification for the different steps and some broader claims are missing. I try to dissect this below more precisely.

Major points:

Based on the materials and methods reader cannot necessarily understand how the editing of phage genomes has been done. Maybe at least add some reference to the methods part. Resistances of each plasmid should be mentioned, it is not enough to mention only the antibiotics used.

How the environmental samples were treated, figure 2 alone does not explain the details.

How the newly modified phages were isolated and sequenced? Also, based on the text it is unclear why the target phage was selected via scarless CRISPR editing in the first place. That procedure either is not well explained in the manuscript. Can you elaborate these?

Also, showing that the tail fibers could be changed in more than one case needs supporting proof. Have all the resulting phages sequenced? Now only the PCR results from the environmental sample are listed. At least it should be mentioned if it was done in other cases.

Row 117: To actually support the claims of the methodology being capable of extending the host range of the PHB20 phage more examples should be shown. Now authors only list one example where one additional strain could be infected by the phage. Data not shwn could be supplementary. Or at least some numbers about how many other phages could infect alternative hosts and how many hosts.

Figure 1C: Please add to the legend what the grey background means

Figure 3C: Nothing is explained about the samples 1-14 in AGE gel, nor the controls.

Additionally in terms of the language, there are some parts I found difficult to understand:

Minor points:

Row 16: Screening directly in water is a bit misleading. I would say isolation source or something specifying that the genes can be obtained from other phages in the same environment. Also, the title of the manuscript should be formulated again in that aspect. I am not sure if it as such is proper sentence. Elsewhere I think it is formulated better and more understandable.

Row 29: Maybe change replace to complement or could be used as an alternative for. I don’t think they can entirely replace, since broad spectrum of antibiotics is despite al the negative effects very useful for example in cases when treatment is needed urgently.

Row 48: I would chance to: 96% of all known phages

Rows 54-55: There is some word missing from this sentence: However, their research only genetically modified publicly published cultivable phage.

Row 62: Change: doing so has successfully expanded the host spectrum of recombinant PHB20 phages. to: doing so successfully expanded the host spectrum of recombinant PHB20 phages.

Row 123: This sentence does not make sense, please modify: The genetically modified phages can meet the current large-scale rapidly  screening in order to screen out effective antibacterial agents.

Row 126: This sentence is a bit too straightforward in my opinion, please modify. Maybe you could suggest that this is one option to meet the needs, now it is kind of suggested as the only solution: To meet the current needs of antibacterial agents, it is necessary to establish recombinant phage libraries in the face of the continuous increase of multidrug-resistant bacteria.

I also want to point out that for E. coli, it is quite easy to find new phages from the environment and many of them are also easy to cultivate, in addition large phage collections of them exist. Maybe authors could treat this as proof of principle and suggest that in the case of other bacteria species this approach could be more beneficial if phages are not easy to cultivate.

Author Response

We would like to express our sincere thank you comments and suggestions, which allowed us to further improve our manuscript. This manuscript has been carefully modified according to the comments. The following response is point-by-point towards the reviewers’ comments.

  1. The manuscript: ‘The strategy of phage tail fiber genes to obtain the source of water environment: Engineering phage for targeted multidrug resistant Escherichia coli’ introduces a gene editing approach to modify narrow host range phage. It is shown that after modification of tail fiber the PHB20 phage can infect alternative host carrying multiple antibiotic resistances. There are several points I want to address and I recommend major modification prior considering for publication.

The authors have an interesting approach to fight increasing antibiotic resistance. I think the research itself has been conducted using novel and valid technique but there are some parts in the manuscript not explained well and missing essential information. I also understand it is a communication article with limited space but mainly the procedure and the justification for the different steps and some broader claims are missing. I try to dissect this below more precisely.

We thank the reviewer for the positive comment. We have modified accordingly.

Major points:

  1. Based on the materials and methods reader cannot necessarily understand how the editing of phage genomes has been done. Maybe at least add some reference to the methods part. Resistances of each plasmid should be mentioned, it is not enough to mention only the antibiotics used.

Thanks for the suggestions. We have add the reference and methods in the manuscript.

  1. How the environmental samples were treated, figure 2 alone does not explain the details.

Thanks for the suggestions. Following the operation shown in Figure 2, we collected 43 environmental water samples, and removed the bacteria.

  1. How the newly modified phages were isolated and sequenced? Also, based on the text it is unclear why the target phage was selected via scarless CRISPR editing in the first place. That procedure either is not well explained in the manuscript. Can you elaborate these?

Thanks for the suggestions. From the results, 30 different phage tail fiber genes were obtained. Subsequently, we constructed 30 of recombinant PHB20 phages, all of which showed potential antibacterial abilities.

We have used this system to edit phages in the early days. “Recently, we used the Scarless Cas9 Assisted Recombineering (no-SCAR) system as a powerful tool for genome editing of an E. coli phage (Chen et al., 2021)”

  1. Also, showing that the tail fibers could be changed in more than one case needs supporting proof. Have all the resulting phages sequenced? Now only the PCR results from the environmental sample are listed. At least it should be mentioned if it was done in other cases.

Thanks for the suggestions. We constructed 30 of recombinant PHB20 phages, all of which showed potential antibacterial abilities.

  1. Row 117: To actually support the claims of the methodology being capable of extending the host range of the PHB20 phage more examples should be shown. Now authors only list one example where one additional strain could be infected by the phage. Data not shwn could be supplementary. Or at least some numbers about how many other phages could infect alternative hosts and how many hosts.

Thanks for the suggestions. We constructed 30 of recombinant PHB20 phages. In fact, these recombinant phage only infected original strain and target strain. Here, we will focus on this approach. In subsequent studies, we found that the recombinant phage only infected specific O-antigen E. coli.

  1. Figure 1C: Please add to the legend what the grey background means

Thanks for the suggestions. (C) The efficiency of plating was determined by plaque assay. This experiment was performed in triplicate. PHB20 (Genbank no. MN481366), PHB13 (Genbank no. MK573636), and SP01 (Genbank no. KY114934). Different symbols represent different protospacer sequences.

  1. Figure 3C:Nothing is explained about the samples 1-14 in AGE gel, nor the controls.

Thanks for the suggestions. (C) Sanger sequencing of the target genes. The numbers 1-14 represent recombinant phages.

Additionally in terms of the language, there are some parts I found difficult to understand:

Minor points:

  1. Row 16:Screening directly in water is a bit misleading. I would say isolation source or something specifying that the genes can be obtained from other phages in the same environment. Also, the title of the manuscript should be formulated again in that aspect. I am not sure if it as such is proper sentence. Elsewhere I think it is formulated better and more understandable.

Thanks for the suggestions. Engineering phage for targeted multidrug-resistant Escherichia coli

  1. Row 29:Maybe change replace to complement or could be used as an alternative for. I don’t think they can entirely replace, since broad spectrum of antibiotics is despite al the negative effects very useful for example in cases when treatment is needed urgently.

Thanks for the suggestions. As viruses that specifically infect bacteria, bacteriophages could be used as alternative to antibiotics in treating bacterial infections in plants, animals, and human beings

  1. Row 48:I would chance to: 96% of all known phages

Thanks for the suggestions. Ninety-six percent of all known phages are tailed.

  1. Rows 54-55:There is some word missing from this sentence: However, their research only genetically modified publicly published cultivable phage.

Thanks for the suggestions. However, most research has on genetically modifying  cultivable phages. In addition, some factors, including operability, time consumption, and recombination efficiency, limit the development of gene-editing technologies.

  1. Row 62:Change: doing so has successfully expanded the host spectrum of recombinant PHB20 phages. to: doing so successfully expanded the host spectrum of recombinant PHB20 phages.

Thanks for the suggestions. We found that doing so successfully expanded the host spectrum of recombinant PHB20 phages.

  1. Row 123:This sentence does not make sense, please modify: The genetically modified phages can meet the current large-scale rapidly  screening in order to screen out effective antibacterial agents.

Thanks for the suggestions. Subsequently, swapping these phage tail fiber genes established an engineered phage library. Similarly, the metagenomic database contains abundant phage tail fiber genes, which can be synthesized and swapped with endogenous genes to form  the desired model phage particles.

  1. Row 126: This sentence is a bit too straightforward in my opinion, please modify. Maybe you could suggest that this is one option to meet the needs, now it is kind of suggested as the only solution:To meet the current needs of antibacterial agents, it is necessary to establish recombinant phage libraries in the face of the continuous increase of multidrug-resistant bacteria.

Thanks for the suggestions. It is possible to employ recombinant phage libraries to meet the current need for new antibacterial agents in the face of a continuous increase in multidrug-resistant bacteria.

I also want to point out that for E. coli, it is quite easy to find new phages from the environment and many of them are also easy to cultivate, in addition large phage collections of them exist. Maybe authors could treat this as proof of principle and suggest that in the case of other bacteria species this approach could be more beneficial if phages are not easy to cultivate.

Thanks for the suggestions.

Round 2

Reviewer 1 Report

I believe that this manuscript can be published in the IJMS.

Reviewer 2 Report

I think after modifications the manuscript is suitable for publication